# Effects of Light Intensity on Endogenous Hormones and Key Enzyme Activities of Anthocyanin Synthesis in Blueberry Leaves

Xiaoli An [1], Tianyu Tan [2], Xinyu Zhang [1], Xiaolan Guo [3], Yunzheng Zhu [1], Zejun Song [1] and Delu Wang [1,*]

1  College of Forestry, Guizhou University, Huaxi, Guiyang 550025, China; anxiaoli61@163.com (X.A.); zhangxinyu_x@163.com (X.Z.); zhuyunzheng04@163.com (Y.Z.); zejun202212@163.com (Z.S.)
2  Forestry Bureau of Kaili, Kaili 556000, China; 18786671054@163.com
3  College of Life Sciences, Huizhou University, Huizhou 516007, China; gxl2022@hzu.edu.cn
*  Correspondence: dlwang@gzu.edu.cn

**Abstract:** Plant anthocyanin is a secondary metabolite widely distributed in the roots, stems, leaves, flowers and fruits of plants, and its synthesis is significantly affected by light intensity. To reveal the physiological response mechanism of anthocyanin synthesis in blueberry leaves at different light intensities, four light intensities (100% (CK), 75%, 50% and 25%) were set for the 'O'Neal' southern highbush blueberry as the experimental material in our study. The relationship between endogenous hormone contents, key enzyme activities, and variations in the anthocyanin content in blueberry leaves under various light intensities during the white fruit stage (S1), purple fruit stage (S2) and blue fruit stage (S3) of fruit development were studied. The results showed that the anthocyanin content of blueberry leaves increased first and then decreased, and decreased first and then increased with the increase in light intensity and development stage, respectively. The appropriate light intensity could significantly promote the synthesis of anthocyanin, and the anthocyanin content in leaves treated with 75% light intensity was 1.09~4.08 times that of other light intensity treatments. The content or activities of gibberellin ($GA_3$), indoleacetic acid (IAA), jasmonic acid (JA), abscisic acid (ABA), ethylene (ETH), phenylalanine ammonia lyase (PAL), chalcone isomerase (CHI), dihydroflavonol reductase (DFR) and UDP-glucose: flavonoid 3-glucosyltransferase (UFGT) were significantly or extremely significantly correlated with the content of anthocyanin in leaves. This indicated that light intensity significantly promoted anthocyanin synthesis in blueberry leaves by affecting endogenous hormone contents and key enzyme activities in the anthocyanin synthesis pathway. This study lays a foundation for further research on the molecular mechanism of light intensity regulating anthocyanin synthesis in blueberry leaves.

**Keywords:** light intensity; blueberry; anthocyanin; endogenous hormones; key enzyme activities

## 1. Introduction

Blueberry (Ericaceae, *Vaccinium*) leaves are rich in various nutrients such as anthocyanins, flavonoids and polyphenols, and they can be used as a sustainable and low-cost plant material to extract anthocyanins. Thus far, over 600 anthocyanins have been identified in nature [1], and anthocyanins in leaves mainly exist in vacuoles of leaf epidermal cells [2,3] or glandular hairs [4], which can be used as antioxidants to protect plants from damage caused by UV radiation [5], freezing and drought stress [6]. Light is one of the key environmental factors affecting anthocyanin synthesis in many plants, among which light intensity is the most significant [7,8].

Phytohormones have been shown to play an important role in regulating plant responses to environmental stress [9], and can participate in the regulation of anthocyanin synthesis [10]. For example, Luo et al. [11] found that genes related to IAA, ABA, ETH, JA and GA in rapeseed seedlings responded to high light stress. Aux (Auxin), ABA, JA

and GA (gibberellic acid) regulate the function and expression of transcriptions factors of the MYB-bHLH-WD40 complex and flavonoid biosynthesis pathway genes involved in the anthocyanin branch [12]. MeJA (methyl jasmonate) and SA (salicylic acid) were both found to stimulate anthocyanin production in the callus cultures of *Daucus carota* [13]. Accumulation of anthocyanin was suppressed by shading in grape berry skins [14]. In Arabidopsis thaliana, strong light can regulate anthocyanin content by stimulating JA content [15].

The biosynthetic pathway of anthocyanins is mainly divided into three stages. Firstly, 4-coumaroyl-CoA is synthesized by the precursor phenylalanine via PAL, C4H (cinnamate-4-hydroxylase) and 4CL (4-Coumarate: CoA ligase) [16]. Secondly, 4-coumaroyl-CoA and malonyl CoA are catalyzed by CHS (Chalcone synthase) to synthesize tetrahydroxychalcone, which is isomerized by CHI to form the colorless compound trihydroxyflavanone, which is then further catalyzed by F3H (flavanone-3-hydroxylase) to synthesize flavanones and dihydroflavonols [17]. Finally, flavanones and dihydroflavonols are catalyzed by DFR to reduce the 4-position of C ring to produce different colorless anthocyanins. These colorless anthocyanins are catalyzed by ANS (anthocyanidin synthase) to produce colored anthocyanins, and UFGT catalyzes the combination of colored anthocyanins with glycosides to transform those into colored anthocyanins [18]. Studies have found that light intensity regulates plant anthocyanin synthesis by inducing the expression of associated genes and enzyme activities in metabolic pathways [19]. Increasing light intensity promoted the expression levels of *MYB*, *CHS* and *F3H* genes of anthocyanin in coleus, thus inducing anthocyanin synthesis and increasing anthocyanin content [20]. Zhu et al. [21] clarified that under low light stress, the activities of CHI, CHS and F3H involved in the anthocyanin biosynthesis pathway of purple cabbage decreased, resulting in a decrease in anthocyanin content.

The molecular mechanism of anthocyanin biosynthesis induced by light intensity has been reported in blueberry [8,22]. However, it is not clear how the light intensity affects the physiological mechanism of anthocyanin content by affecting the endogenous hormone contents and the key enzyme activities in the anthocyanin synthesis pathway. Therefore, this study analyzed the changes in anthocyanin content in blueberry leaves under different light intensities at stage S1, S2 and S3, and its correlation with the content of endogenous hormones ($GA_3$, JA, IAA, ABA and ETH) and the activities of key enzymes (PAL, CHI, DFR and UFGT). We sought to explore the correlation between anthocyanin content, endogenous hormone contents and key activities under different light intensities, and to analyze the synergistic regulation of anthocyanin biosynthesis by light intensity, hormones and key enzymes from the physiological level, to provide a scientific basis for the control of light intensity in blueberry production.

## 2. Materials and Methods

### 2.1. Overview of the Experimental Site

The experimental site is located in the Experimental Nursery of the College of Forestry, South Campus of Guizhou University, Huaxi District, Guiyang, with an altitude of 1159 m, 104°34′ east longitude, and 26°34′ north latitude. It is a subtropical humid and moderate climate. The maximum temperature is 39.5 °C, the minimum temperature is −9.5 °C and the average annual temperature is 15.8 °C. The yearly effective accumulated temperature above 10 °C is 4637.5 °C, the annual precipitation is 1229 mm, the annual average relative humidity is 79% and the total integrated solar radiation is 3567 MJ/m$^2$.

### 2.2. Experimental Materials

Four-year-old southern highbush blueberry variety 'O'Neal' with the same maturity and growth was used as the experimental material, and the test seedlings were transplanted into plastic flower pots (inner diameter 26.5 cm, bottom diameter 17.5 cm, height 19.7 cm). One seedling per pot was cultured with pine forest humus as the substrate. The nutrient

content of the substrate is high, with a pH of about 4.8, which can satisfy the normal growth of blueberries, and weeding and irrigation were carried out regularly.

### 2.3. Experimental Design

As shown in Table 1, the four light intensities were 100% (CK group, natural light), 75% (light shading), 50% (moderate shading) and 25% (severe shading) full light intensity, which were controlled by an illuminance meter (Shenzhen Jumaoyuan Technology Co., Ltd., Shenzhen, China) and black sunshade nets of different densities with 2, 3, 4, 6 and 8 needles. Three replicates were set for each treatment, with 10 plants in each group. The light intensity was measured at three random locations under the sunshade nets. The trial began after the blueberries had bloomed (1 April 2020).

**Table 1.** Actual light intensity corresponding to relative light intensity.

| Light Intensity | S1/$\mu mol \cdot m^{-2} s^{-1}$ | S2/$\mu mol \cdot m^{-2} s^{-1}$ | S3/$\mu mol \cdot m^{-2} s^{-1}$ |
|---|---|---|---|
| 25% | 372 ± 34.06 Ad | 369 ± 29.44 Ad | 379 ± 28.29 Ad |
| 50% | 750 ± 31.18 Ac | 699 ± 24.83 Ac | 778 ± 30.02 Ac |
| 75% | 1123 ± 40.99 Ab | 1094 ± 48.50 Ab | 1143 ± 36.37 Ab |
| CK | 1498 ± 39.26 Aa | 1456 ± 44.46 Aa | 1587 ± 37.53 Aa |

Note: The above table shows the light intensity at 10 a.m. as measured with a photometer. S1: white fruit stage, S2: purple fruit stage, S3: blue fruit stage. In the table, different uppercase letters indicate significant differences in the same light intensity during different stages, and different lowercase letters indicate significant differences in different light intensity treatments at the same stage ($p < 0.05$); values represent mean ± standard error.

### 2.4. Sample Collection

After one month of treatment, according to the test scheme, blueberry plants with consistent growth were selected for sample collection. The healthy leaves with the same size were randomly sampled in the group at three fruit development stages of 28 days (white fruit stage, S1), 35 days (purple fruit stage, S2) and 42 days (blue fruit stage, S3) after full bloom. We sampled 10 g from each biological replicate in each stage, and a total of 3 biological replicates were used for experimental research. The samples were placed in a screw-tip-bottom centrifuge tube wrapped with tin foil paper, stored in liquid nitrogen and returned to the laboratory for storage in an ultra-low temperature refrigerator at −80 °C.

### 2.5. Method of Index Determination

2.5.1. Methods for Determination of Endogenous Hormone Contents and Enzyme Activities

The contents of endogenous hormones and the activities of key enzymes in the anthocyanin synthesis pathway were determined by double antibody sandwich enzyme-linked immunosorbent assay (ELISA) [23]. The collected blueberry leaves were tested using an ELISA kit produced by Guizhou Wela Technology Limited Liability Company. Sample treatment: The tissue was rinsed with pre-cooled PBS (0.01 M, pH = 7.4), and the weighed 0.1 g leaf and the corresponding volume of PBS (according to the weight to volume ratio of 1:9) were added to the homogenizer for grinding. To further lyse the tissue cells, the homogenate was broken by ultrasound. Finally, the homogenate was centrifuged at 10,000 rpm for 5 min, and the supernatant was taken for detection. The content of gibberellin 3 ($GA_3$), jasmonic acid (JA), indoleacetic acid (IAA), abscisic acid (ABA) and ethylene (ETH), and the activities of phenylalanine ammonia lyase (PAL), chalcone isomerase (CHI), dihydroflavonol reductase (DFR) and UDP-glucose: flavonoid 3-glucosyltransferase (UFGT) were detected.

2.5.2. Method for Determination of Anthocyanin Content

The anthocyanin content of blueberry leaves was determined using a Solarbio biochemical kit. (1) We weighed and ground 0.1g of blueberry leaf samples with a low-temperature grinding machine. In order to further lyse tissue cells, appropriate ultrasonic fragmentation was performed. We added 1 mL of the extract, and it was transferred to the EP tube after being fully homogenized. The extract was diluted to 1 mL, covered and extracted at 60 °C

for 40 min, during which time it was high-speed shocked 8 times. We then centrifuged at 12,000 rpm, held at 4 °C for 15 min and took the supernatant for testing. (2) The microplate reader was preheated for 30 min, recalibrated and the wavelength was adjusted. (3) We added 40 μL of samples to the determination tubes (1.5 mL EP tube) 1 and 2, and then we added 160 μL of reagent 1 and reagent 2, respectively. (4) After mixing, we centrifuged at 10,000 rpm for 15 min, placed 150 μL of supernatant in 96-well plates and detected its absorbance. The absorbance values of tube 1 at 530 nm and 700 nm were recorded as A1 and A1′, respectively, and the absorbance values of tube 2 at 530 nm and 700 nm were recorded as A2 and A2′, respectively.

The anthocyanin content was calculated according to Formulas (1) and (2).

$$\Delta A = (A1 - A1') - (A2 - A2') \tag{1}$$

Anthocyanin content (μmol/gFW):

$$[\Delta A \div (\varepsilon \times d) \times 103 \times F] \times V \div W = 0.31 \times \Delta A \div W \tag{2}$$

where $\varepsilon$: molar extinction coefficient, $2.69 \times 10^4$ mL/mmol/cm; d: 96-hole plate optical path length, 0.6 cm; 103: 1 mmol = 103 μmol; F: dilution multiple, 5; V: total volume of extract, 1 mL; W: fresh weight of sample, g.

*2.6. Data Analysis*

Excel 2019 and Origin 2022 were used for sorting, calculating, mapping data and correlation analysis. One-way ANOVA and Tukey's test were performed using SPSS 19.0. Statistical differences were marked by sequential letter labeling.

## 3. Results

*3.1. Effects of Light Intensity on Endogenous Hormone Contents in Blueberry Leaves*

As shown in Figure 1, the content of five endogenous hormones in blueberry leaves was significantly affected by light intensity and development stage. Except for IAA at S3, the content of $GA_3$ and IAA in other treatments decreased gradually with the increase in light intensity and leaf development, while the content of JA, ABA and ETH increased gradually with the increase in light intensity and leaf development.

The effects of light intensity and development stage on the content of $GA_3$ and IAA in blueberry leaves are shown in Figure 1a,b. Under the same light intensity treatment, the contents of $GA_3$ and IAA at S1 were significantly higher than those at S2 and S3. The contents of $GA_3$ and IAA at 25% light intensity treatment were significantly higher than those under other light intensity treatments during the same development stage, while the content of IAA at S3 was the opposite. Under 25% light intensity treatment, the $GA_3$ content in leaves of S1 was 13.94%, 11.30% and 6.68% higher than that of CK, 75% and 50% light intensity treatments at the same stage, while S2 was 3.04%, 3.27% and 1.76% higher, and S3 was 17.07%, 8.62% and 2.29% higher, respectively. From S1 to S3, the content of IAA in leaves treated with 25% light intensity was 1.04~1.07 times, 1.03~1.48 times and 0.54~0.91 times that of other light intensity treatments during the same stage, respectively.

The effects of light intensity and developmental stage on the content of JA, ABA and ETH in blueberry leaves are shown in Figure 1c–e. Under the same light intensity, the contents of JA, ABA and ETH in leaves at S3 were significantly higher than those at S1 and S2. From S1 to S3, the contents of JA, ABA and ETH in leaves treated with 25% light intensity were significantly lower than those treated with other light intensities at the same stage. Compared with CK, the JA content of leaves during S1 decreased by 6.78%~52.85%, with an average decrease of 28.00%; the decrease rate at S2 was 4.70%~28.12%, with the average decrease rate was 14.83%; and the decrease at S3 was 2.62%~15.15%, with an average decrease of 7.56%. The ABA content under CK treatment at S3 was as high as 127.10 ng/g, which was 1.06~1.2 times that of other light intensity treatments at the same stage, and 1.78 and 1.16 times that of the same light intensity treatment at S1 and S2,

respectively. The content of ETH in leaves was less, but it also had a similar change rule with the content of JA and ABA. Among them, the content of ETH under CK treatment at S3 was the highest, which was 46.71 μg/g, that is, 1.02~1.18 times that of other light intensity treatments during the same stage. The results showed that full light and late development were more conducive to the synthesis of JA, ABA and ETH content in leaves, but the increase rate of these hormone contents gradually decreased with the continuous development of leaves.

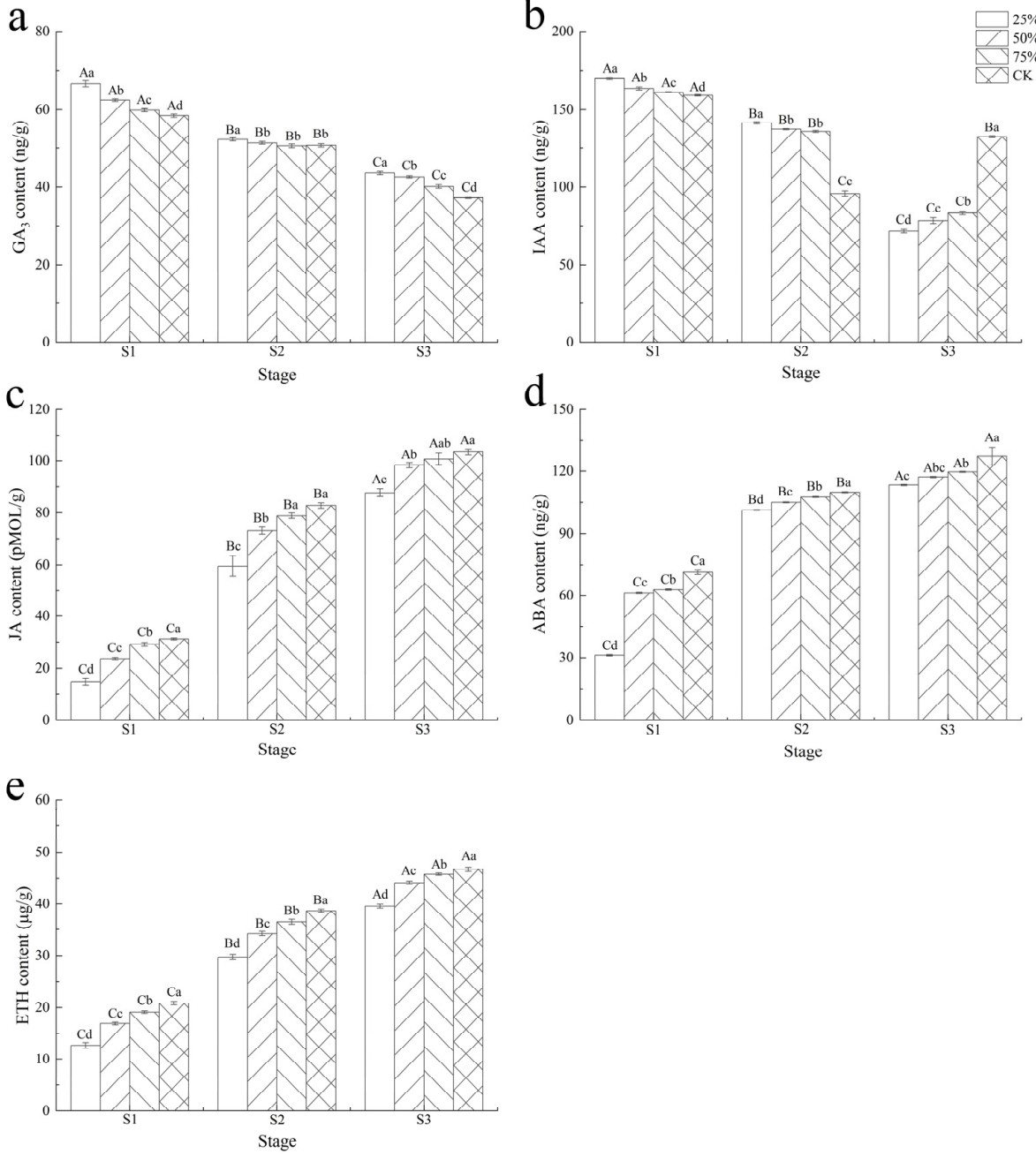

**Figure 1.** Effects of light intensity on endogenous hormone (GA$_3$ (**a**), IAA (**b**), JA (**c**), ABA (**d**) and ETH (**e**)) contents in blueberry leaves. Note: In the figure, different uppercase letters indicate significant differences at the same light intensity during different stages, and different lowercase letters indicate significant differences under different light intensity treatments during the same stage ($p < 0.05$). Error bars represent SD.

### 3.2. Effects of Light Intensity on Key Enzyme Activities in the Anthocyanin Synthesis Pathway of Blueberry Leaves

The effects of different light intensities and development stages on the activities of key enzymes in the anthocyanin synthesis pathway in blueberry leaves are shown in Figure 2. The activities of PAL, CHI, DFR and UFGT in leaves were significantly affected by light intensity and development stage. These four enzyme activities gradually increased with the increase in light intensity and leaf development, but the extent of the increase gradually decreased.

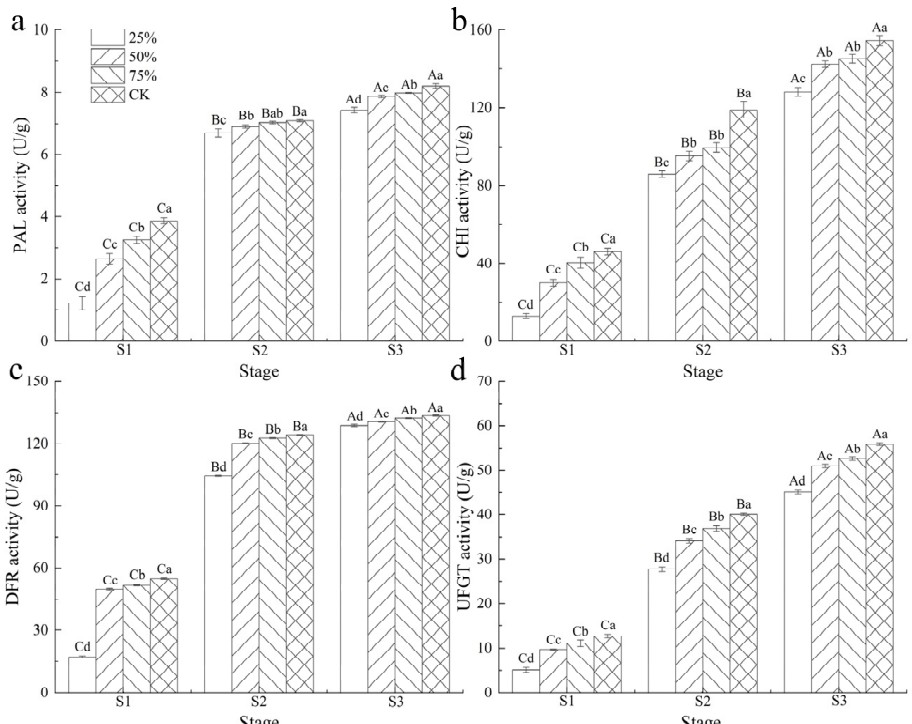

**Figure 2.** Effects of light intensity on key enzyme activities (PAL (**a**), CHI (**b**), DFR (**c**) and UFGT (**d**)) in the anthocyanin synthesis pathway of blueberry leaves. Note: In the figure, different uppercase letters indicate significant differences at the same light intensity during different stages, and different lowercase letters indicate significant differences under different light intensity treatments during the same stage ($p < 0.05$). Error bars represent SD.

Under the same light intensity, the activities of four enzymes in leaves at S3 were significantly higher than those of S1 and S2, and the four enzyme activities of CK treatment at each stage were significantly higher than those of shading treatment at the same stage. Compared with the other three key enzymes, the PAL enzyme activity in blueberry leaves was the lowest, which was 1.22 U/g under 25% light intensity at S1, and 8.20 U/g under CK treatment at S3. Compared with CK, the CHI activity of leaves at S1 decreased by 12.22%~72.03%, with an average decrease of 39.60%. At S2, it decreased by 16.28%~27.83%, with an average decrease of 21.32%, and at S3, it decreased by 6.01%~17.03%, with an average decrease of 10.31%. The activities of DFR and UFGT in leaves under different light intensities also had the same variation pattern at different developmental stages. It can be seen that the four enzyme activities in the anthocyanin synthesis pathway of blueberry leaves are positively correlated with light intensity and development stage.

### 3.3. Effect of Light Intensity on Anthocyanin Content in Blueberry Leaves

As shown in Figure 3, light intensity and development stage had significant effects on the anthocyanin content in blueberry leaves, and the anthocyanin content at S3 was significantly higher than that at S1 and S2 under the same light intensity treatment. At

the same development stage, the anthocyanin content in leaves increased first and then decreased with the increase in light intensity, and reached the peak at 75% light intensity. Under the same light intensity treatment, the anthocyanin content in leaves treated with 25% light intensity gradually increased with leaf development, while the anthocyanin content in leaves treated with the other three light intensities decreased first and then increased with leaf development, and the decrease was smaller than the increase. The anthocyanin content of leaves treated with CK and 75% light intensity at S1 and S3 was significantly higher than that of the other two treatments, while the anthocyanin content of leaves treated with 75% light intensity at S2 was significantly higher than that of the other three treatments. The anthocyanin content of leaves under 75% light intensity at S3 was as high as 1.122 μmol/gFW, which was 1.19 times, 1.76 times and 2.45 times those of CK, 50% and 25% light intensity treatments at the same stage, and 1.42 times and 1.47 times those of 75% light intensity treatment at S1 and S2, respectively. This indicated that too low or too high light intensity was not conducive to the synthesis of anthocyanin in blueberry leaves, and 75% light intensity was more conducive to the synthesis of anthocyanin in leaves.

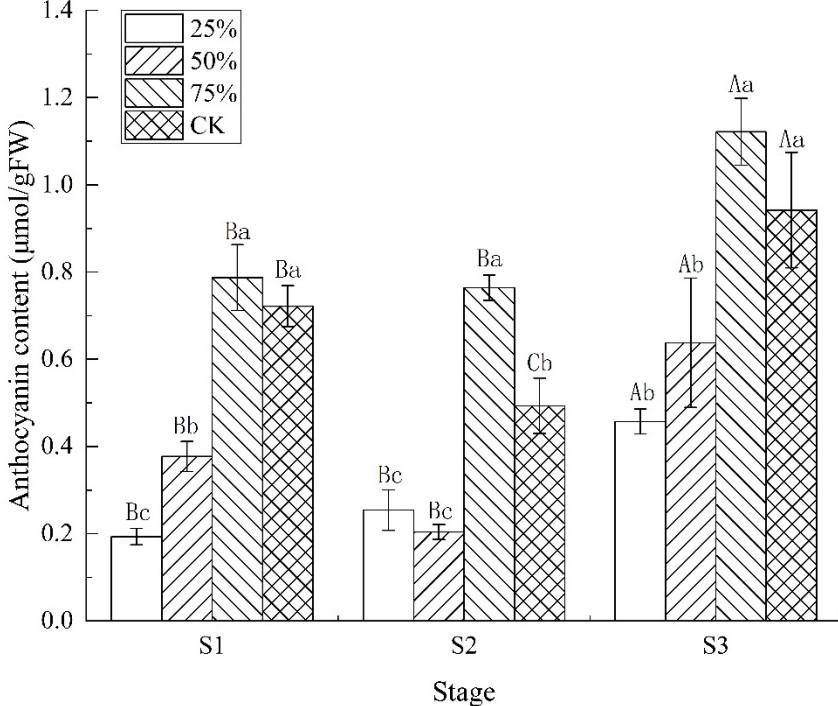

**Figure 3.** Effect of light intensity on anthocyanin content in blueberry leaves. Note: In the figure, different uppercase letters indicate significant differences at the same light intensity during different stages, and different lowercase letters indicate significant differences under different light intensity treatments during the same stage (*p* < 0.05). Error bars represent SD.

### 3.4. Correlation Analysis

The correlation analysis of anthocyanin content with light intensity, endogenous hormones (GA$_3$, JA, IAA, ABA, ETH) and the activities of key enzymes in the anthocyanin synthesis pathway (PAL, CHI, DFR, UFGT) in blueberry leaves at different developmental stages under different light intensity conditions is shown in Figure 4. From S1 to S2, the content of GA$_3$ and IAA in leaves had an extremely significant negative correlation with light intensity, but at S3, the former had an extremely significant negative correlation and the latter had an extremely significant positive correlation. The anthocyanin content, other three hormones and four enzyme activities at the three stages were extremely significantly or significantly positively correlated with light intensity. This was consistent with the trend of changes in anthocyanin content, endogenous hormone contents and key enzyme activities in the anthocyanin synthesis pathway with light intensity, as mentioned above.

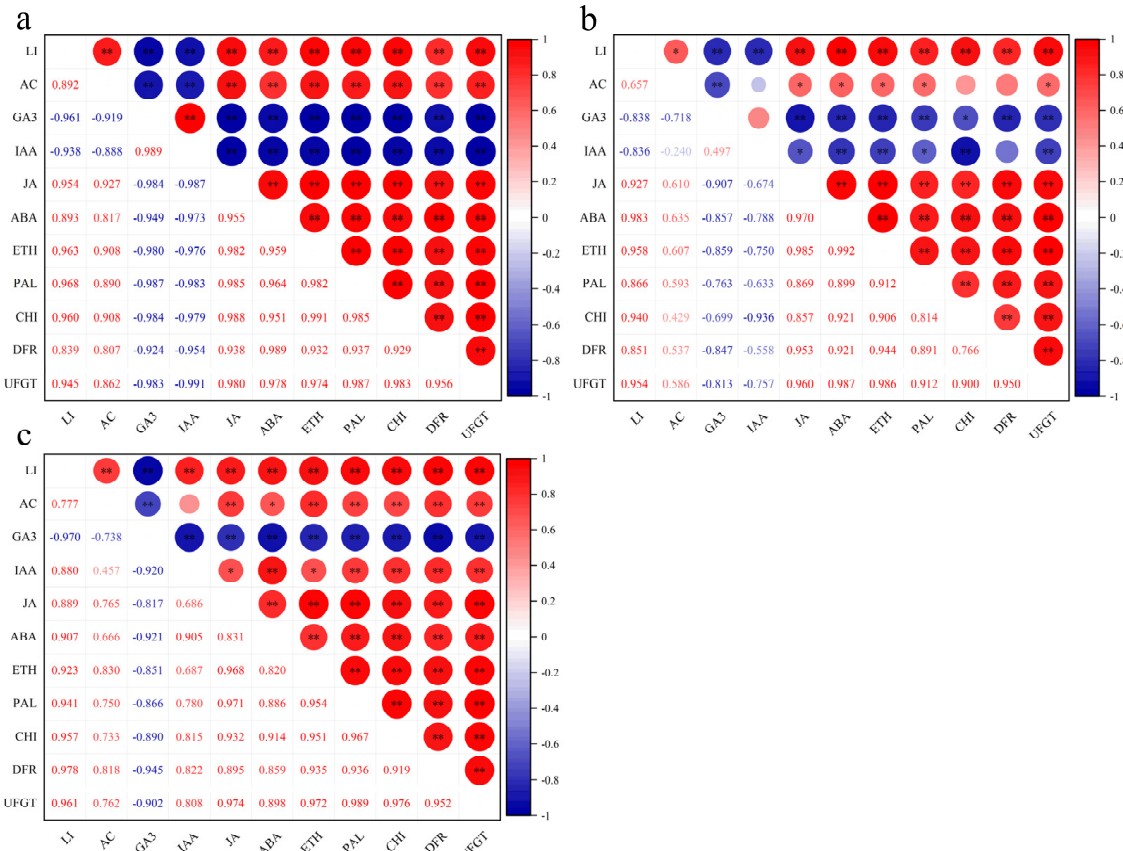

**Figure 4.** Correlation analysis of anthocyanin content in blueberry leaves with light intensity, endogenous hormones and enzyme activities. Note: LI: light intensity, AC: anthocyanin content; (**a**): white fruit stage (S1), (**b**): purple fruit stage (S2), (**c**): blue fruit stage (S3); * $p \leq 0.05$, significant correlation; ** $p \leq 0.01$, extremely significant correlation.

From S1 to S3, the anthocyanin content in leaves treated with different light intensities showed an extremely significant negative correlation with GA$_3$ content, and an extremely significant negative correlation with IAA content at first, then a negative correlation and then a positive correlation. The anthocyanin content of leaves treated with different light intensities had an extremely significant or significant positive correlation with the other three hormone contents and four enzyme activities at S1 and S2, and had a significant positive correlation with the content of JA, ABA and ETH, and the activities of PAL and UFGT at S2. At the same time, from S1 to S3, the correlation coefficients between the contents of five hormones, the activities of four enzymes and anthocyanin content decreased first and then increased, which was similar to the trend of anthocyanin content changing with light intensity.

In addition, light intensity, anthocyanin content, five hormone contents and four enzyme activities were significantly correlated with any two factors at S1. There was no significant correlation between anthocyanin content and the IAA content, CHI activity and DFR activity, and between IAA content and the GA$_3$ content and DFR activity, at S2; there was no significant correlation between anthocyanin content and the IAA content at S3; and there was significant correlation between any other two factors. From S1 to S3, the content of JA, ABA and ETH and the activities of PAL, CHI, DFR and UFGT in leaves had a significant or extremely significant negative correlation with GA$_3$ content, and showed an extremely significant negative correlation with IAA content at first, though that then turned to a negative correlation, and then to a significant positive correlation. However, the IAA content was first significantly positively correlated with GA$_3$ content, then positively correlated, and then significantly negatively correlated. The results showed that the light

intensity's regulation of anthocyanin synthesis in blueberry leaves was closely related to five endogenous hormones and four key enzyme activities in the anthocyanin synthesis pathway, and there was also a certain relationship between hormones and enzyme activities.

It follows that the anthocyanin content, endogenous hormone contents and key enzyme activities in the anthocyanin synthesis pathway in blueberry leaves are significantly correlated with light intensity and development stage. However, the correlations of some factors at different development stages and at different light intensities are quite different.

## 4. Discussion

### 4.1. Light Intensity Promotes Anthocyanin Synthesis by Regulating the Content of Endogenous Hormones

Hormones affect all aspects of plant development and growth physiology, including the biosynthesis of anthocyanin [24], and the prerequisite for inducing anthocyanin synthesis in vegetative tissues is light [2]. In the leaves of oilseed peony, light, moderate and severe shadings decreased the ABA concentration by 8.8%, 14.4% and 22.7% but increased the IAA concentration by 38.1%, 45.5% and 49.0% and the $GA_3$ concentration by 6.3%, 7.6% and 11.7%, respectively [25]. The $GA_3$ content in *Carpinus betulus* L. seedlings increased with the decrease in light intensity, and the IAA content decreased with the decrease in light intensity [26]. In the two studies cited, the IAA contents showed different trends with light intensity, but they were consistent with the IAA content trend at S1, S2 and S3 in this study. At the same time, they were consistent with the results in this study in that the IAA content in blueberry leaves and light intensity showed a highly significant negative correlation at S1 and S2, and a highly significant positive correlation at S3. It has been reported that high light intensity triggers the biosynthesis of ABA, which, in turn, promotes the expression of anthocyanin biosynthetic genes and enhances anthocyanin biosynthesis [27]. Exogenous ethylene treatment markedly improved the expression levels of the anthocyanin-biosynthesis-related genes (*PsPAL*, *PsDRF*, *PsANS*, *PsUFGT*, etc.) in plum, thus accelerating anthocyanin accumulation [28]. JA promoted anthocyanin biosynthesis in leaves of rapeseed seedlings [11] and apple [29]. Exogenous ABA treatment enhanced anthocyanin accumulation in grape berry skins [14]. Plant growth regulators ABA and ethephon promoted anthocyanin synthesis in chicory (*Cichorium intybus* L.), while $GA_3$ inhibited its anthocyanin synthesis [30]. In summary, the literature indicates that light intensity can promote the synthesis of anthocyanins by regulating the biosynthesis of hormones.

During the whole process of blueberry fruit development, the anthocyanin content was highly significantly positively correlated with the ABA and ETH content, and highly significantly negatively correlated with the IAA content [31]. The $GA_4$ content was strongly negatively correlated with the anthocyanin content in sweet cherry [32]. There was a strong positive correlation between the ABA content and anthocyanin content in *Lycium* fruit [33] and purple-leaved cultivars of tea [34]. The IAA content was positively correlated with the anthocyanin content during bicolor leaf development [35]. In our study, the anthocyanin content was significantly or extremely significantly positively correlated with the ETH and ABA contents, and extremely significantly negatively correlated with the $GA_3$ content, which supported the above views. However, from S1 to S3, the IAA content and anthocyanin content were extremely significantly negatively correlated, negatively correlated and positively correlated, respectively. The differences in findings between studies may have been caused by differences in plant species, sampling organs and leaf growth and development. Nonetheless, the contents of JA and ETH were strongly positively correlated with the anthocyanin content in our study, which was confirmed in *Saxifraga longifolia* leaves [36] and plum fruits [28]. This indicates that the synergistic effect of different hormones promotes the biosynthesis of anthocyanin.

*4.2. Light Intensity Regulates Anthocyanin Synthesis by Inducing the Expression of Enzyme Activities*

The activities of four enzymes (PAL, CHI, DFR, UFGT) in this study showed an upward trend with the increase in light intensity and leaf development, and the enzyme activities at three development stages were significantly positively correlated with light intensity. It has been reported that light intensity can induce the expression of CHS, CHI, F3′5′H and DFR genes in the peel of sand pear [37] and grape [38], thus significantly inducing anthocyanin accumulation. That supports the proposal that the expression of enzyme genes in the anthocyanin synthesis pathway can regulate the level of enzyme activities to a certain extent, which regulates the content of anthocyanin, a proposal that is consistent with the results of this study. In *Perilla frutescens* var. *crispa*, the PAL activity under low light intensity was lower than that under normal light intensity [39]. DFR activity in 'Fuji' apple peel increased with the increase in light intensity, and anthocyanin synthesis was regulated by DFR activity [40]. The lack of anthocyanin accumulation in Matthiola line white flowers [41] and ivy [42] is due to a lack of DFR activity, while the lack of anthocyanin in white grapes [43] is due to a lack of UFGT activity. The results indicate that light intensity can affect anthocyanin synthesis by regulating the activity of key enzymes in the anthocyanin biosynthesis pathway through the signal transduction pathway.

The anthocyanin content in eggplant was significantly positively correlated with the expression levels of *SmCHI* and *SmDFR* [44], while the DFR activity was significantly correlated with anthocyanin accumulation in apple [45]. Meanwhile, the PAL activity was strongly positively correlated with anthocyanin biosynthesis in eggplant peel and fruit [46], which was consistent with the results of this study. At the same time, there was no significant relationship between CHI and DFR activities and the anthocyanin content at S2 in this study, which may be related to the decrease in anthocyanin biosynthesis and the increase in anthocyanin degradation at S2. Combined with the study of enzyme activities under different light intensity conditions, the comparative analysis of its variation law confirmed that the light intensity will affect the biosynthesis of anthocyanin by inducing key enzyme activities.

*4.3. Regulation of Light Intensity in Anthocyanin Biosynthesis*

Generally speaking, inducing the synthesis of anthocyanin requires high light intensity, and the anthocyanin content in plant leaves is related to light levels [2]. Manetas [47] pointed out that anthocyanins are ubiquitous in green leaves, but their content is not well covered by the color of chlorophyll, meaning it is not easy to detect. Later, the absorption value of anthocyanin was instead detected in green leaves, which confirmed the presence of anthocyanin in green leaves [48]. The blueberry leaf samples collected in this study were all green leaves with low anthocyanin content, but they were significantly affected by light intensity and leaf development. Research has found that the concentration of anthocyanin in grapes gradually decreases with a decrease in light transmittance [49]. Meanwhile, strong light (100% light transmittance) inhibited anthocyanin synthesis in *Petunia* corollas [50]. Elsewhere, the anthocyanin content in leaves of four subtropical dominant tree species gradually decreased with their growth and development, and shading (30% light transmittance) inhibited the accumulation of anthocyanin in leaves [51]. These results are consistent with the results of this study, indicating that anthocyanin synthesis of blueberry leaves is closely related to light intensity and leaf development. In addition, the anthocyanin content in the leaves of the three stages reached its highest under 75% light intensity treatment, indicating that excessive light intensity may reduce anthocyanin biosynthesis or increase anthocyanin degradation, leading to a decrease in anthocyanin content in the leaves.

**5. Conclusions**

Light is a fundamental requirement for plant growth and development, but excessive light intensity can cause irreversible damage to chloroplasts and cell metabolism. Correlation analysis showed that light intensity, endogenous hormones and key enzyme activities

had significant effects on anthocyanin content in blueberry leaves. Among them, the content of GA$_3$ was negatively correlated with the content of anthocyanin in all three stages. This study showed that light intensity affected anthocyanin synthesis by regulating the content of endogenous hormones in blueberry leaves and the activities of key enzymes in the anthocyanin synthesis pathway, and 75% light intensity was the most conducive to anthocyanin biosynthesis in blueberry leaves. Blueberry leaves are byproducts with potential economic value, and these findings help us understand the potential mechanisms by which light intensity regulates anthocyanin synthesis and accumulation in blueberry leaves.

**Author Contributions:** Data curation, visualization and writing—original draft, X.A.; investigation, T.T.; data curation, X.Z.; formal analysis, X.G., Y.Z. and Z.S.; resources, writing—review and editing, funding acquisition and supervision, D.W. All authors have read and agreed to the published version of the manuscript.

**Funding:** This work was supported by the National Natural Science Foundation of China (grant number 31760205).

**Institutional Review Board Statement:** Not applicable.

**Informed Consent Statement:** Not applicable.

**Data Availability Statement:** Not applicable.

**Conflicts of Interest:** The authors declare no conflict of interest.

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
