# Peer review of "Effects of Light Intensity on Endogenous Hormones and Key Enzyme Activities of Anthocyanin Synthesis in Blueberry Leaves"

_horticulturae, doi:10.3390/horticulturae9060618_

Round 1

Reviewer 1 Report

The Manuscript "Effects of light intensity on endogenous hormones and key enzyme activities of anthocyanin synthesis in blueberry leaves" is interesting to read and represents an excellent basis for one of the future researches as a comparison for the application of artificial light.

In present form there are some minor spelling correction needed and addition clarification as follows:

In section 1 at line 44 “GA” mentioned for the first time to add meaning or it is a spelling correction needed and should be written “GA3

at line 139 “The ultrasonic wave was appropriately broken „ It is necessary to clarify purpose for which the ultrasound was used, it is not mentioned anywhere else in the manuscript add additional information for equipment used and described the process of ultrasound application.

In present form there are some minor spelling correction needed.

Reviewer 2 Report

The MS under review is aimed to reveal how light intensity controls the content of anthocyanins in blueberry leaves. Unfortunately the MS has many shortcomings. 

First of all, it is not clrear from the introduction why should we expect that anthocyanin synthesis in blueberry leaves is affected by light intensity not like in other plans and requires special investigation. No general information on the dependence of anthocyanin content on light intensity and developmental stage is pesent. No information why blueberry leaves require special attention in this respect was provided.  

Secondly, many (or even most) plant characteristics wether increase or decrease with the changes in light intensity. If particular characteristics (for instance, content of one of the endogenuous hormone and the content of anthocyanins) change in the same way (for instance, both increase) it does not nesesserily mean that one process (for example, anthosyanin synthesis) depends on the other characteristic. They may change the same way being independent. There is no information that all basic hormones are involved in the regulation of anthocyanin synthesis. Thus, the conclusions are not convincing based on the results of correlation analysis only. 

Minor concerns:

Line 19-20: ... decreased first and increased with the developmental stage. Decreased relatively to what?

Line 101: Please, provide the information on the manufacturer of the photometer. 

Fig. 1b. Are there suggested explanations of the IAA decrease in S2 at CK and increase in S3 at CK. It doesn't seem to have logical explanation. 

Lines 319-320. The conclusion is not based on the results of the work. 

Lines 381--383: It can not be concluded from the results of this work wether the decrease in anthocyanin content at 100% light intensity is caused by decreases synthesis or increased degradation.  

Round 2

Reviewer 2 Report

Some questions are answered, but some require repetition of the experiment to be sure that decreases/increases are not accidental and have logical explanation. 

Author Response

Response to Reviewer 2 comments

Point 1: Some questions are answered, but some require repetition of the experiment to be sure that decreases/increases are not accidental and have logical explanation.

Response 1: Thank you for your careful review and helpful comments and suggestions on our manuscript. These suggestions are also of great significance to our future research. Wish you a happy life!